# Volatile Compounds of Sucuk, a Dry Fermented Sausage: The Effects of Ripening Rate, Autochthonous Starter Cultures and Fat Type

**DOI:** 10.3390/foods13233839

**Published:** 2024-11-28

**Authors:** Mükerrem Kaya, Güzin Kaban

**Affiliations:** Department of Food Engineering, Faculty of Agriculture, Atatürk University, 25240 Erzurum, Türkiye

**Keywords:** fermented sausage, sucuk, volatile compounds, starter culture, sheep tail fat

## Abstract

The aim of this study was to determine the effects of ripening rate (slow or fast), usage autochthonous starter cultures (control—spontaneous fermentation, *Lactiplantibacillus plantarum* GM77, *Staphylococcus xylosus* GM92 or *L. plantarum* GM77 + *S. xylosus* GM92) and type of fat (beef fat-BF, sheep tail fat-STF and BF+STF) on the volatile compounds of sucuk (a Turkish dry fermented sausage). A total of 74 volatile compounds were identified, including groups of aliphatic hydrocarbons, aldehydes, ketones, alcohols, sulfide compounds, esters, aromatic hydrocarbons, nitrogenous compounds, acids and terpenes in sucuk. Slow ripening resulted in significant increases in the abundance of ethanol, acetic acid, ethyl acetate, acetoin and diacetyl. A similar situation was determined for a mixed culture (*L. plantarum* + *S. xylosus)*. Correlation analysis showed that the effects of slow ripening and mixed culture use were more pronounced in terms of volatile compound content. Although the effect of fat type on volatile compounds was quite limited compared to other factors, correlation analysis showed that STF had a different volatile compound profile.

## 1. Introduction

Aroma compounds that are effective in shaping consumers’ food preferences generally consist of volatile organic compounds [1]. These compounds can be found in foods as a result of physiological and/or enzymatic processes, and can also be formed by chemical, biochemical or microbial changes during the production and storage of foods [2]. Fresh meat has little aroma and only a blood-like taste [3,4]. However, meat products produced using one or more of these processes, such as fermentation, drying and curing, have different flavor profiles, and each meat product has its own characteristic aroma.

Among meat products, fermented meat products (fermented sausage and ripened meat products) stand out with their unique sensory properties. In these products, many volatile compounds are formed as a result of reactions such as carbohydrate fermentation, lipolysis, proteolysis, lipid oxidation and amino acid catabolism. In addition, spices included in the formulation during production are also an important source of volatile compounds [5]. Aliphatic hydrocarbons, aldehydes, acids, ketones, alcohols, sulfur compounds, esters, aromatic hydrocarbons and terpenes are volatile compounds commonly found in fermented meat products. Non-volatile compounds such as amino acids and peptides are also effective in the formation of the typical flavor of fermented sausages [6]. There are fermented meat products specific to each geography. The first information about fermented sausages dates back to times B.C. [7]. The first information about sucuk, a Turkish dry fermented sausage, is in a work called Divânü Lugâti’t-Türk, the oldest known Turkish dictionary, which was written in 1072 [8,9]. Sucuk is the only dry fermented sausage in Türkiye. Since this product has a high share in meat products, it is also important from an economic perspective. Sucuk is also produced industrially today and starter cultures are often used in its production. However, it is not possible to provide the unique taste and aroma of sausage using commercial culture preparations. Therefore, research is being conducted on the possibility of using autochthonous strains as starter cultures in sucuk production. A few studies have been conducted in this context [10,11,12,13].

The rate of ripening is of great importance in terms of the volatile compounds that result from the degradation of proteins and lipids [14]. In addition, the release of volatile compounds from the matrix can also be affected by the drying process [15]. Two different ripening rates are applied in sucuk production: slow and fast. Therefore, initial fermentation temperatures ranging from 12 to 26 °C are used and the production period varies between 6 and 20 days [10,16]. In a study carried out on sucuk, it was found that fast ripening in the presence of starter culture resulted in a lower water activity value and an increased a* value. It was also reported that the use of sheep tail fat in the production of this dry fermented sausage type did not affect the instrumental color values. In the same study, it was also found that the use of mixed cultures (*Lactiplantibacillus plantarum* + *Staphylococcus xylosus*) increased the overall acceptability scores for both slow and fast ripening [13].

Fat is important in fermented sausages for sensory properties such as firmness, juiciness, mouthfeel and flavor, as well as for technological functions [17,18,19]. Lipolysis and lipid oxidation play an important role in the flavor development of dry fermented sausages. The free fatty acids that are formed as a result of lipolysis are further subjected to lipid oxidation, resulting in the formation of a large number of volatile compounds [18]. In sucuk production, unlike many other fermented sausages, sheep tail fat (STF) can also be included in the formulation [20]. STF, which has a higher unsaturated fatty acid content than meat fat, is more sensitive to lipid oxidation [13] Akköse et al. [13] reported that the use of STF in sucuk increases the TBARS value, which is an important measure of the extent of lipid oxidation. However, no research has been found on the effect of using STF in sausage on volatile compounds. The aim of this study was to investigate the effects of the usage of autochthonous strains (spontaneous fermentation, *Lactiplantibacillus plantarum* GM77, *Staphylococcus xylosus* GM92 or *L. plantarum* GM77/*S. xylosus* GM92), type of fat (beef fat-BF, STF and BF+STF) and ripening rate (slow or fast) on the volatile profile of sucuk product, a dry fermented sausage type.

## 2. Materials and Methods

### 2.1. Material

Beef meat from the shoulder of carcasses (conditioned at 4 °C for 24 h) was obtained from a local slaughterhouse (Meat and Dairy Institution, Erzurum, Türkiye). Both beef fat and sheep tail fat were used in the study, and were also taken from the same slaughterhouse.

### 2.2. Production of Sucuk

Sucuk production was carried out as a parallel research project to that of Akköse et al. [13]. In the preparation of sucuk batters, 80% lean meat and 20% fat (20% beef fat-BF, 20% sheep tail fat-STF or 10% BF + 10% STF) were used. For 1 kg of meat–fat mixture (4:1), 25 g of salt, 10 g of garlic, 4 g of sucrose, 2.5 g of pimento, 9 g of cumin, 7 g of red pepper, and 5 g of black pepper were used. Nitrate (150 mg/kg KNO_3_) was added to the batters for the slow ripening, and nitrite (150 mg/kg NaNO_2_) for the fast ripening. *Lactiplantibacillus plantarum* GM77 (about 10^7^ cfu/g), *Staphylococcus xylosus* GM92 (about 10^6^ cfu/g) and *L. plantarum* GM77 + *S. xylosus* GM92 were used as starter cultures [13]. As a control, batters were prepared without the starter culture. Two batches were produced for each treatment, resulting in forty-eight batches according to the experimental design (Table 1).

Sucuk batters were prepared using a laboratory bowl cutter (MADO type MTK 662, Dornhan, Schwarzwald, Germany). After filling the prepared batters into collagen casings (38 mm, Naturin Darm, Germany) with a laboratory-type filling machine (MTK 591, Mado, Dornhan, Schwarzwald), the samples were placed in a climatic chamber (Reich, Urbach, Germany) with automatic temperature and humidity control. According to the experimental design, the slow ripening program was continued at 18 ± 1 °C between days 1 and 10 and at 16 ± 1 °C between days 11 and 14; the fast ripening program was started at 24 ± 1 °C on day 1, then moved to 22 ± 1 °C for days 2 and 3, to 20 ± 1 °C for days 4 and 6, to 18 ± 1 °C for days 7 and 10 and to 16 ± 1 °C for days 11 and 14. The relative humidity was gradually reduced from 92 ± 2% to 84 ± 2% during both ripening periods.

### 2.3. Volatile Compounds Analysis

A solid-phase microextraction method was used for the extraction of volatile compounds from sucuk samples. After the sample (5 g) was placed in a 40 mL vial for extraction, it was placed in a thermal block. CAR/PDMS fiber (Supelco, Bellefonte, PA, USA) was inserted into the vial, which was kept at 30 °C for 1 h in the thermal block, and the volatile compounds in the headspace were collected for 2 h at the same temperature. Then, the fiber was injected into a GC/MS device (Gas Chromatography/Mass Spectrometry, Agilent Technologies 6890N/Agilent Technologies 5973, Agilent, Santa Clara, CA, USA). The oven temperature program was started from 40 °C (5 min) and gradually increased, and when the temperature reached 210 °C, it was kept for 12 min. DB-624 (30 m, 0.25 mm id, 1.4 µm film thickness, J&W Scientific, Folsom, CA, USA) was used in the identification. The mobile phase in the system was helium and the flow rate was 1 mL/min. The obtained results were determined by comparing them with the mass spectrometry library (NIST, Wiley and Flavor) and using standard substances. Additionally, the Kovats index was calculated using a standard mix (Supelco 44585-U, Bellefonte, PA, USA) in the definition. The results were expressed as arbitrary area units (×10^6^) [11].

### 2.4. Statistical Analysis

The study was conducted taking ripening rate, use of autochthonous starter cultures and the type of fat into consideration. The trial was carried out using a completely randomized design with two replicates (two batters for each treatment). Ripening rate, starter culture, and type of fat were evaluated as the main effects, and replications were taken as the random effects. The means of significant sources of variation were compared by Duncan’s multiple range tests (SPSS, IBM Inc., Chicago, IL, USA). The results were expressed as the mean value ± standard deviation. Cluster analysis of a heat map was also performed using the chiplot program to determine the relationship between factors and volatile compounds or chemical groups of volatile compounds. In the algorithm, complete was taken as the method and correlation was taken as the distance (https://www.chiplot.online, accessed on 15 October 2024).

## 3. Results and Discussion

A total of 74 volatile compounds, including the aliphatic hydrocarbons, aldehydes, ketones, alcohols, sulfide compounds, esters, aromatic hydrocarbons, nitrogenous compounds, acid and terpenes groups were identified (Table 2, Table 3, Table 4 and Table 5).

### 3.1. Aldehydes

A total of seven aldehydes were determined in sucuk groups. Acetaldehyde was only influenced by the ripening rate and fast ripening gave a higher average value than slow ripening. For pentanal, slow ripening showed a higher abundance than fast ripening. Similarly, Stahnke [21] reported higher levels of lipid oxidation products in fast ripening (high fermentation temperature, glucose, nitrite and *Pediococcus pentosaceus*) than in slow ripening (nitrate and low fermentation temperature). The use of sheep tail fat (STF) increased the abundance of this compound formed by lipid oxidation. Furthermore, pentanal level increased in the presence of *L. plantarum* GM77 + *S. xylosus* GM92. The lowest level of 2-methyl-3-phenylpropanal was observed when *L. plantarum* GM77 was used as a starter culture (Table 2). *L. plantarum* was also reported in another study to reduce the abundance of this compound [12]. The primary markers of lipid oxidation, such as hexanal, pentanal and nonanal, are the saturated aldehydes and decadienals, whose intensities increase as oxidation progresses [22]. In the present study, hexanal, octanal, heptanal and nonanal were not affected by fat type, starter culture and ripening rate (Table 2). Aldehydes can also be formed by amino acid deamination or transamination, Strecker degradation and microbial activity during fermentation [2].

### 3.2. Acids

In the sucuk samples, acetic acid was determined (Table 2). Acetic acid can be produced by homofermentative lactic acid bacteria and staphylococci, as well as by fatty acid oxidation and alanine catabolism [23,24]. Slow ripening resulted in higher mean values than fast ripening. Some differences were also observed between fat groups; the highest mean value was found in sheep tail fat, but this mean value was not statistically different from the mean value of beef fat (Table 2). The *L. plantarum* GM77 + *S. xylosus* GM92 mixed culture gave significantly higher mean values than the other groups, including the control group. Similarly, an increase in acetic acid content was reported in sucuk with this mixed culture [11].

### 3.3. Ketones

Three ketone compounds were determined: diacetyl, acetone and acetoin. The ripening rate and starter culture were effective on acetone and acetoin, while all factors were effective on diacetyl (Table 2). Diacetyl and acetoin were determined in quite high amounts for slow ripening (Table 2). Both compounds were determined in considerable amounts in the presence of *L. plantarum* GM77 + *S. xylosus* GM92 (Table 2). Zheng et al. [25] also reported that *L. plantarum* YR07 and *Mammaliicoccus sciuri* S.18 greatly promoted the formation of acetoin (3-hydroxy-2-butanone). In addition, it was also indicated that the interactions between the strains used are an important factor in the formation of volatile compounds [25,26]. Acetoin and diacetyl are low-molecular-weight compounds that are formed during the fermentation process. The formation of such specific compounds during carbohydrate fermentation depends on the starter culture used [27,28]. Indeed, the breakdown of pyruvic and lactic acid by lactic acid bacteria produces compounds such as diacetyl, acetoin [29], acetic acid and ethanol, which are responsible for specific flavors [30]. Ketones can derive from lipid oxidation, citrate and glucose metabolism [2], amino acid degradation and microbial metabolism [23]. Ketones usually provide fruity or musty notes, while some smaller ketones (such as acetoin and diacetyl) exhibit a buttery flavor [2].

### 3.4. Nitrogenous Compounds

1-methyl-1H-pyrrole was affected by all the factors examined. It showed higher mean values for slow ripening, STF usage and the *L. plantarum* GM77 + *S. xylosus* GM92 group (Table 2). Pyrroles can result in the pyrolysis of amino acids, the reaction of ammonia with dicarbonyls, or the interaction of furfurals and ammonia [3], and can also arise from the production of Maillard reaction products in the presence of lipid oxidation products [31]. On the other hand, microbial metabolism is very important when it comes to fermented sausage aroma, and microbial degradation of amino acids can be a source of aroma compounds such as straight-chain sulfur compounds, thiols, pyrazines and pyrroles [30].

**Table 2 foods-13-03839-t002:** The overall effects of ripening rate, fat type and starter culture on aldehydes, acids, nitrogenous compounds and ketones of sucuk (mean ± standard deviation) (Au×10^6^).

Compounds	KI	RI	Ripening Rate		Fat Type			Starter Culture			
			Slow	Fast	BF	STF	BF+STF	Control	Lp	Sx	Lp+Sx
Aldehydes											
Acetaldehyde	<500	a	1.80 ± 4.59 ^b^	7.00 ± 8.87 ^a^	3.82 ± 7.78	4.88 ± 8.89	4.51 ± 5.60	4.39 ± 5.35	6.93 ± 11.76	3.32 ± 5.44	2.98 ± 4.89
Pentanal	742	a	1.57 ± 3.81 ^a^	0.34 ± 0.71 ^b^	0.22 ± 0.82 ^b^	1.93 ± 4.03 ^a^	0.71 ± 2.33 ^a^	0.33 ± 0.75 ^b^	0.65 ± 0.97 ^b^	0.12 ± 0.33 ^b^	2.70 ± 5.11 ^a^
Hexanal	849	a	10.10 ± 14.20	7.76 ± 36.93	10.00 ± 35.80	9.06 ± 16.10	7.74 ± 28.73	9.87 ± 32.39	10.10 ± 41.05	3.09 ± 3.34	12.68 ± 19.58
Heptanal	955	a	1.27 ± 3.24	1.67 ± 11.76	2.79 ± 14.50	0.85 ± 2.65	0.76 ± 2.25	0.50 ± 0.58	3.37 ± 16.58	0.31 ± 0.41	1.69 ± 4.54
Octanal	1044	a	0.47 ± 2.57	0.15 ± 0.62	0.07 ± 0.24	0.71 ± 3.20	0.14 ± 0.36	0.03 ± 0.20	0.51 ± 0.93	0.00 ± 0.00	0.69 ± 3.61
Nonanal	1146	a	1.43 ± 8.40	0.51 ± 1.25	0.38 ± 0.98	0.31 ± 0.79	2.23 ± 10.27	0.00 ± 0.00	1.07 ± 1.34	0.00 ± 0.00	2.81 ± 11.83
2-methyl-3-phenyl propanal	1318	b	46.97 ± 42.85	39.51 ± 45.14	42.85 ± 33.50	44.26 ± 44.97	42.62 ± 52.42	55.02 ± 59.46 ^a^	18.12 ± 6.81 ^b^	42.50 ± 18.04 ^a^	57.32 ± 54.88 ^a^
Acids											
Acetic acid	717	a	118.37 ± 221.73 ^a^	11.37 ± 9.11 ^b^	69.65 ± 166.42 ^a^	93.66 ± 208.40 ^a^	31.30 ± 100.07 ^b^	14.04 ± 13.40 ^b^	20.70 ± 10.12 ^b^	14.21 ± 8.04 ^b^	210.54 ± 286.70 ^a^
Ketones											
Acetone	541	a	0.00 ± 0.00 ^b^	0.47 ± 0.83 ^a^	0.30 ± 0.92	0.23 ± 0.49	0.18 ± 0.35	0.00 ± 0.00 ^c^	0.64 ± 1.10 ^a^	0.00 ± 0.00 ^c^	0.30 ± 0.37 ^b^
Diacetyl	645	a	9.85 ± 20.21 ^a^	0.95 ± 1.85 ^b^	6.18 ± 15.02 ^a^	8.69 ± 20.59 ^a^	1.34 ± 2.26 ^b^	0.04 ± 0.21 ^c^	5.67 ± 4.76 ^b^	0.00 ± 0.00 ^c^	15.91 ± 26.86 ^a^
Acetoin	779	a	7.40 ± 15.19 ^a^	0.11 ± 0.49 ^b^	4.06 ± 10.97	4.72 ± 13.36	2.48 ± 9.40	0.00 ± 0.00 ^b^	2.74 ± 3.02 ^b^	0.00 ± 0.00 ^b^	12.28 ± 20.23 ^a^
Nitrogenous compounds											
1-methyl-1H-pyrrole	786	b	2.30 ± 4.79 ^a^	0.58 ± 0.75 ^b^	1.27 ± 2.85 ^b^	2.19 ± 4.77 ^a^	0.86 ± 2.43 ^b^	0.39 ± 1.07 ^b^	0.90 ± 0.93 ^b^	0.33 ± 0.39 ^b^	4.15 ± 6.19 ^a^

BF: beef fat; STF: sheep tail fat; Lp: *L. plantarum* GM77; Sx: *S. xylosus* GM92; KI: Kovats index calculated for DB-624 column installed on GC/MS; RI: reliability of identification; a: mass spectrum and retention time identical with authentic sample; b: mass spectrum and Kovats index from literature in accordance. ^a–c^: Means marked with different letters in same row in same factor are statistically different (*p* < 0.05).

### 3.5. Aromatic and Aliphatic Hydrocarbons

Seven aromatic hydrocarbons were determined in sucuk samples. As seen in Table 3, generally higher abundances were determined for slow ripening compared to fast ripening. The type of fat had a significant effect on two compounds, p-xylene and styrene. In addition, the *L. plantarum* GM77+*S. xylosus* group generally showed higher abundance than the other groups (Table 3). The source of aromatic hydrocarbons varies considerably. For example, it has been shown that toluene can be originated from lipid degradation, grasses used as animal feed or amino acid catabolism [32,33].

Among the aliphatic hydrocarbons, undecane showed a high abundance. This compound was not affected by ripening rate and fat type. However, use of starter culture had a significant effect on undecane; the mixture culture showed higher abundance than the control and *L. plantarum* GM77. Slow ripening caused an increase in the content of heptane. Among fat groups, the STF group showed low abundance compared to the BF and BF+STF groups (Table 3). Yılmaz Oral and Kaban [34] also determined seven aliphatic hydrocarbons in heat-treated sucuk and reported that only undecane was affected by the use of starter culture among these compounds. In another study, it was reported that heptane and tridecane were not affected by the use of starter culture, and other determined aliphatic hydrocarbons increased in the case of starter culture use. In the same study, they reported that toluene, p-xylene and styrene levels decreased when starter culture was used [35].

**Table 3 foods-13-03839-t003:** Overall effect of ripening rate, fat type and starter culture on aromatic hydrocarbon and aliphatic hydrocarbon of sucuk (mean ± standard deviation) (Au×10^6^).

Compounds	KI	RI	Ripening Rate		Fat Type			Starter Culture			
			Slow	Fast	BF	STF	BF+STF	Control	Lp	Sx	Lp+Sx
Aromatic hydrocarbons											
Toluene	785	a	6.47 ± 8.94 ^a^	1.69 ± 1.59 ^b^	4.73 ± 9.16	4.27 ± 6.47	3.25 ± 3.85	5.89 ± 9.88 ^a^	1.97 ± 1.36 ^b^	2.52 ± 2.03 ^b^	5.94 ± 8.57 ^a^
p-xylene	892	a	1.42 ± 2.82 ^a^	0.59 ± 1.09 ^b^	0.30 ± 0.62 ^b^	1.93 ± 3.24 ^a^	0.79 ± 1.41 ^b^	0.59 ± 0.71 ^b^	0.49 ± 0.65 ^b^	0.99 ± 1.45 ^b^	1.95 ± 3.85 ^a^
Styrene	916	b	0.49 ± 1.37	0.30 ± 0.54	0.33 ± 0.65 ^b^	0.70 ± 1.63 ^a^	0.15 ± 0.32 ^b^	0.23 ± 0.35	0.31 ± 0.71	0.34 ± 0.51	0.69 ± 1.86
1-methyl-4-(1-methylethyl)-benzene	1060	b	216.09 ± 264.57 ^a^	63.95 ± 58.86 ^b^	160.19 ± 239.90	132.74 ± 190.97	127.13 ± 184.51	105.70 ± 57.36 ^b^	88.40 ± 63.51 ^b^	106.15 ± 131.13 ^b^	259.82 ± 358.35 ^a^
1-methyl-4-(1-methyl ethenyl)-benzene	1112	b	10.44 ± 13.72 ^a^	2.71 ± 2.52 ^b^	5.48 ± 7.41	7.53 ± 10.81	6.72 ± 12.88	4.52 ± 4.19 ^b^	4.90 ± 4.43 ^b^	4.47 ± 3.96 ^b^	12.42 ± 18.88 ^a^
1-methoxy-4-(1-propenyl)-benzene	1342	b	0.10 ± 0.27 ^b^	0.35 ± 0.44 ^a^	0.25 ± 0.37	0.17 ± 0.27	0.27 ± 0.49	0.16 ± 0.28 ^b^	0.15 ± 0.39 ^b^	0.34 ± 0.39 ^a^	0.25 ± 0.45 ^ab^
1,2-dimethoxy-4-(2-propenyl)-benzene	1457	b	6.30 ± 9.45 ^a^	4.18 ± 2.26 ^b^	4.93 ± 5.45	6.05 ± 8.43	4.75 ± 6.66	2.83 ± 2.74 ^b^	3.25 ± 2.09 ^b^	3.10 ± 2.02 ^b^	11.77 ± 11.03 ^a^
Aliphatic hydrocarbons											
Heptane	700	a	13.89 ± 28.19 ^a^	1.16 ± 2.21 ^b^	15.20 ± 33.16 ^a^	3.05 ± 4.43 ^b^	4.32 ± 11.03 ^a^	14.24 ± 34.45 ^a^	1.61 ± 2.14 ^b^	8.02 ± 14.71 ^ab^	6.23 ± 17.21 ^ab^
Nonane	900	a	0.13 ± 0.44	0.06 ± 0.23	0.07 ± 0.26	0.09 ± 0.33	0.13 ± 0.45	0.27 ± 0.60 ^a^	0.00 ± 0.00 ^a^	0.11 ± 0.32 ^b^	0.00 ± 0.00 ^b^
Decane	1000	a	1.04 ± 2.35	0.65 ± 181	0.83 ± 2.18	1.11 ± 2.58	0.61 ± 1.40	0.75 ± 1.87 ^ab^	0.37 ± 1.05 ^b^	1.60 ± 2.78 ^a^	0.67 ± 2.21 ^b^
Undecane	1100	a	15.55 ± 33.13	10.64 ± 7.54	11.31 ± 16.12	15.65 ± 30.61	12.32 ± 23.57	7.56 ± 7.30 ^b^	7.28 ± 7.13 ^b^	14.80 ± 10.53 ^ab^	22.75 ± 44.57 ^a^
Dodecane	1200	a	2.96 ± 10.55 ^a^	0.38 ± 0.89 ^b^	1.92 ± 5.68	0.75 ± 1.46	2.34 ± 11.78	4.30 ± 14.60	0.41 ± 0.56	1.20 ± 2.39	0.77 ± 2.25
Tridecane	1300	a	0.47 ± 0.66 ^a^	0.14 ± 0.36 ^b^	0.26 ± 0.52	0.31 ± 0.62	0.35 ± 0.53	0.55 ± 0.67 ^a^	0.20 ± 0.43 ^bc^	0.39 ± 0.57 ^ab^	0.07 ± 0.42 ^c^
Tetradecane	1400	a	0.55 ± 1.55 ^a^	0.06 ± 0.19 ^b^	0.32 ± 1.43	0.37 ± 1.25	0.22 ± 0.49	0.30 ± 0.71	0.11 ± 0.38	0.21 ± 0.33	0.60 ± 2.07

BF: beef fat; STF: sheep tail fat; Lp: *L. plantarum* GM77; Sx: *S. xylosus* GM92; KI: Kovats index calculated for DB-624 column installed on GC/MS; RI: reliability of identification; a: mass spectrum and retention time identical with authentic sample; b: mass spectrum and Kovats index from literature in accordance. ^a–c^: Means marked with different letters in same row in same factor are statistically different (*p* < 0.05).

### 3.6. Alcohols

Among the alcohols determined in sucuk samples, ethyl alcohol showed the highest abundance. Slow ripening increased ethyl alcohol formation. Fat type had no significant effect on this compound. On the other hand, the starter culture factor had a significant effect on ethyl alcohol and the highest abundance was observed in the presence of *L. plantarum* GM77+*S. xylosus* GM92. (Table 4). Kaban et al. [12] emphasized that the use of starter culture (mono or mixed culture) in sucuk reduces the amount of ethanol and this result is due to the conversion of ethanol to the corresponding acids, aldehydes and esters. On the other hand, Sallan et al. [35] stated that the use of starter culture in heat-treated sucuk had no effect on ethanol. Lipid oxidation, carbohydrate metabolism, methyl ketone reduction and amino acid catabolism are accepted sources of alcohol in dry fermented meat products [2,36,37]. Saturated alcohols, which have relatively higher thresholds, have a limited impact on flavor, whereas unsaturated alcohols, which have lower thresholds, are quite effective regarding flavor [36]. On the other hand, alcohols also play an important role in flavor formation, as precursors of aldehydes and ketones [37].

### 3.7. Sulfide Compounds

Nine sulfide compounds were identified in the sucuk samples. Methyl thiirane, allyl methyl sulfide, 3,3-thiobis-1-propene, methly-2-propenyl disulfide and di-2-propenyl disulfide were the most abundant compounds of this group. Slow ripening significantly increased the abundance of methyl thiirane, allyl methyl sulfide, 3,3-thiobis-1-propene and methly-2-propenyl disulfide (Table 4). On the other hand, methyl thiirane, allyl methyl sulfide and 3,3-thiobis-1-propene showed higher mean abundances in the presence of *L. plantarum* GM77+*S. xylosus* GM92 compared to the other groups (control, *S. xylosus* GM92 and *L. plantarum* GM77). Kargozari et al. [38] reported that methyl allyl trisulfide, allyl mercaptan (2-propene-1-thiol) and diallyl disulfide were significantly affected by the use of *L. plantarum* strains in sucuk production. In addition, high inoculation levels of *S. carnosus* significantly increased the abundances of sulfide compounds, as reported by Tjener et al. [39]. In contrast, in a study on a traditional Spanish dry fermented sausage, it was determined that allyl methyl sulfide, diallyl sulfide and diallyl disulfide were not affected by the use of starter culture [40]. It is also stated that the metabolic complementation between strains causes an increase in the formation of some volatile compounds or in the abundance of volatile compounds [41]. In the present study, only the use of STF increased methyl thiirane abundance. None of the investigated factors showed a significant effect on di-2-propenyl disulfide (Table 4). Since sulfide compounds are highly volatile and have very low perception threshold values, they are effective in the sensory properties of meat products [23].

### 3.8. Esters

Seven esters were determined in sucuk groups. Some of these esters were also determined in previous studies on sucuk [11,12]. As seen in Table 4, ethyl acetate and ethyl 2,4-hexadienoate had a significant abundance among esters. Rapid ripening significantly reduced the amount of both ethyl acetate and ethyl 2,4-hexadienoate. When the starter culture was taken into consideration, the highest abundance for both compounds was observed in the presence of mixed culture (Table 4). In fermented meat products, esters are generally formed as a result of esterification between carboxylic acids and alcohols derived from carbohydrate fermentation or amino acid catabolism [42], and some of them show perception thresholds approximately ten times lower than those of the corresponding alcohols [2]. Esters are important compounds for the aroma of fermented sausages, and they give off a fruity and floral aroma [43,44], and mask rancid odors [35]. The esterase activity of staphylococci is also effective in the formation of esters [15]. On the other hand, Li et al. [45] reported *Staphylococcus* have high esterase activity, and the presence of esterase and acetic acid promotes the formation of esters. In addition, *L. plantarum* strains were reported to enhance the types and relative abundance of esters in Chinese sausages [46]. On the other hand, Kaban et al. [12] found that *L. sakei* as a monoculture in sucuk showed high ester activity. In our study, the fat factor showed an effect on ethyl acetate and the highest average abundance was obtained when only STF was used (Table 4).

**Table 4 foods-13-03839-t004:** Overall effect of ripening rate, fat type and starter culture on alcohols, sulfide compounds and esters of sucuk (mean ± standard deviation) (Au×10^6^).

Compounds	KI	RI	Ripening Rate		Fat Type			Starter Culture			
			Slow	Fast	BF	STF	BF+STF	Control	Lp	Sx	Lp+Sx
Alcohols											
Ethanol	539	a	75.14 ± 137.49 ^a^	35.92 ± 33.10 ^b^	50.67 ± 67.32	54.89 ± 79.77	61.04 ± 142.97	41.45 ± 44.56 ^b^	22.69 ± 27.19 ^b^	25.47 ± 6.63 ^b^	132.51 ± 176.20 ^a^
Isoamyl alcohol	781	b	0.34 ± 1.55	0.13 ± 0.57	0.37 ± 1.86	0.20 ± 0.73	0.14 ± 0.40	0.60 ± 2.16	0.00 ± 0.00	0.35 ± 0.81	0.00 ± 0.00
1-hexacosanol	1097	c	0.00 ± 0.00 ^b^	0.34 ± 0.81 ^a^	0.25 ± 0.63	0.10 ± 0.50	0.17 ± 0.65	0.11 ± 0.50	0.36 ± 0.95	0.11 ± 0.37	0.10 ± 0.36
2-ethyl-1-dodecanol	1102	c	0.00 ± 0.00 ^b^	0.26 ± 0.59 ^a^	0.15 ± 0.48	0.19 ± 0.49	0.05 ± 0.31	0.05 ± 0.21 ^b^	0.19 ± 0.52 ^ab^	0.02 ± 0.13 ^b^	0.25 ± 0.64 ^a^
α–methylbenzyl alcohol	1342	b	1.00 ± 2.60	0.71 ± 0.70	0.61 ± 0.90	1.20 ± 2.75	0.74 ± 1.59	0.50 ± 0.68 ^b^	0.32 ± 0.61 ^b^	0.49 ± 0.56 ^b^	2.09 ± 3.40 ^a^
α–propylbenzene methanol	1357	b	0.00 ± 0.00 ^b^	0.45 ± 1.86 ^a^	0.10 ± 0.47	0.39 ± 2.19	0.19 ± 0.57	0.00 ± 0.00	0.69 ± 2.55	0.06 ± 0.29	0.16 ± 0.58
Sulfide compounds											
Carbon disulfide	552	b	0.00 ± 0.00 ^b^	0.31 ± 0.44 ^a^	0.13 ± 0.28	0.19 ± 0.39	0.14 ± 0.35	0.11 ± 0.28	0.12 ± 0.28	0.22 ± 0.41	0.17 ± 0.40
Methyl thiirane	598	b	237.96 ± 532.07 ^a^	22.80 ± 1.76 ^b^	95.89 ± 311.36 ^b^	215.73 ± 526.70 ^a^	79.53 ± 279.60 ^b^	6.12 ± 5.56 ^b^	33.15 ± 19.09 ^b^	6.99 ± 2.29 ^b^	475.28 ± 677.38 ^a^
Allyl methyl sulfide	730	b	54.23 ± 75.58 ^a^	13.27 ± 11.23 ^b^	36.82 ± 60.94 ^a^	39.64 ± 68.87 ^a^	24.79 ± 39.02 ^b^	24.00 ± 23.45 ^b^	15.16 ± 9.32 ^b^	16.05 ± 8.79 ^b^	79.78 ± 99.50 ^a^
1-(methylthio)-1-propene	753	b	0.00 ± 0.00 ^b^	0.37 ± 1.42 ^a^	0.33 ± 1.60	0.13 ± 0.63	0.09 ± 0.37	0.39 ± 1.83	0.30 ± 0.83	0.05 ± 0.30	0.00 ± 0.00
Dimethyl disulfide	764	b	0.30 ± 0.84	0.51 ± 0.88	0.23 ± 0.89	0.42 ± 0.95	0.55 ± 0.71	0.71 ± 1.13 ^a^	0.01 ± 0.07 ^b^	0.89 ± 1.05 ^a^	0.00 ± 0.00 ^b^
3,3-thiobis-1-propene	888	b	44.30 ± 63.63 ^a^	8.03 ± 8.04 ^b^	28.38 ± 50.65	29.92 ± 56.27	20.18 ± 37.87	16.76 ± 15.10 ^b^	11.44 ± 7.32 ^b^	8.94 ± 6.24 ^b^	67.51 ± 83.67 ^a^
Methly-2-propenyl disulfide	946	b	13.88 ± 13.83 ^a^	9.48 ± 8.82 ^b^	10.84 ± 13.57	12.48 ± 11.16	11.72 ± 1.54	16.25 ± 14.75 ^a^	6.31 ± 4.26 ^c^	13.56 ± 9.13 ^ab^	10.58 ± 13.80 ^b^
Methyl trans-propenyl disulfide	955	b	0.15 ± 0.34 ^b^	0.47 ± 0.58 ^a^	0.23 ± 0.43 ^b^	0.28 ± 0.40 ^b^	0.42 ± 0.63 ^a^	0.61 ± 0.70 ^a^	0.04 ± 0.14 ^b^	0.57 ± 0.42 ^a^	0.02 ± 0.07 ^b^
Di-2-propenyl disulfide	1126	b	33.25 ± 54.50	20.96 ± 17.98	26.34 ± 40.95	26.18 ± 43.28	28.79 ± 39.20	28.86 ± 26.11	21.10 ± 13.84	19.89 ± 7.92	38.55 ± 75.31
Esters											
Ethyl acetate	648	a	17.43 ± 24.36 ^a^	8.68 ± 6.97 ^b^	9.33 ± 11.45 ^b^	17.91 ± 25.02 ^a^	11.93 ± 15.21 ^b^	11.44 ± 8.56 ^b^	5.55 ± 6.07 ^c^	9.44 ± 5.47 ^bc^	25.79 ± 31.62 ^a^
Ethyl butanoate	791	b	0.17 ± 0.49 ^b^	0.52 ± 0.79 ^a^	0.08 ± 0.20 ^b^	0.40 ± 0.74 ^a^	0.55 ± 0.84 ^a^	0.48 ± 0.68 ^a^	0.10 ± 0.42 ^b^	0.59 ± 0.85 ^a^	0.21 ± 0.62 ^b^
Ethyl lactate	843	b	0.14 ± 0.55	0.22 ± 0.46	0.17 ± 0.64	0.20 ± 0.45	0.17 ± 0.40	0.03 ± 0.16	0.26 ± 0.48	0.21 ± 0.47	0.21 ± 0.73
Ethyl 3-methyl butyrate	869	b	0.15 ± 0.44 ^b^	0.49 ± 1.41 ^a^	0.07 ± 0.41 ^b^	0.55 ± 1.59 ^a^	0.33 ± 0.76 ^ab^	0.20 ± 0.57 ^b^	0.01 ± 0.03 ^b^	1.06 ± 1.85 ^a^	0.01 ± 0.03 ^b^
Ethyl 2,4-hexadienoate	1130	c	7.20 ± 15.74 ^a^	0.31 ± 0.84 ^b^	0.94 ± 2.14 ^b^	5.83 ± 12.10 ^a^	4.51 ± 15.73 ^ab^	2.16 ± 2.30 ^b^	0.96 ± 1.19 ^b^	2.00 ± 3.03 ^b^	9.92 ± 22.00 ^a^
Ethyl octanoate	1209	b	0.00 ± 0.00 ^b^	0.42 ± 0.47 ^a^	0.06 ± 0.19 ^c^	0.22 ± 0.33 ^b^	0.34 ± 0.53 ^a^	0.37 ± 0.45 ^a^	0.17 ± 0.35 ^b^	0.11 ± 0.23 ^b^	0.18 ± 0.45 ^b^
Ethyl decanoate	1415	c	0.00 ± 0.00 ^b^	0.07 ± 0.17 ^a^	0.00 ± 0.00 ^b^	0.10 ± 0.19 ^a^	0.01 ± 0.04 ^b^	0.06 ± 0.16 ^ab^	0.01 ± 0.06 ^c^	0.02 ± 0.07 ^bc^	0.07 ± 0.16 ^a^

BF: beef fat; STF: sheep tail fat; Lp: *L. plantarum* GM77; Sx: *S. xylosus* GM92; KI: Kovats index calculated for DB-624 column installed on GC/MS; RI: reliability of identification; a: mass spectrum and retention time identical with authentic sample; b: mass spectrum and Kovats index from literature in accordance; c: tentative identification by mass spectrum. ^a–c^: Means marked with different letters in same row in same factor are statistically different (*p* < 0.05).

### 3.9. Terpenes

A total of 26 terpene compounds were determined in sucuk samples (Table 5). A significant portion of these compounds were also reported in previous studies on sucuk [10,11,12]. Among terpenes, α-pinene, β-myrcene, β-pinene, β-myrcene, α-phellandrene, 3-carene, D-limonene, β-phellandrene, γ-terpinene, linalool, cumic alcohol and caryophyllene were the most abundant compounds of this group, and all of these compounds gave higher abundance in slow ripening than in fast ripening. The type of fat had less effect on terpene compounds. When BF+STF was used, a decrease in 3-carene levels was observed. Similar results were also observed for D-limonene (Table 5). The starter culture factor was effective on many compounds. Copaene, α-pinene, β-pinene, β-myrcene, 3-carene, α- terpinene, D-limonene, β-phellandrene, γ-terpinene, linalool, α –terpineol, cumic alcohol, iso-caryophyllene, caryophyllene and α- caryophyllene showed high abundance in the presence of *L. plantarum* GM77 + *S. xylosus* GM92 mixed culture (Table 5). The source of a significant part of these compounds is spices [47,48,49]. Also, terpenes can result from animal feedstuffs [47]. However, Yılmaz Oral and Kaban [34] reported that *L. sakei* and the *L. sakei* + *S. xylosus* combination increased limonene levels in heat-treated sucuk. In the same study, caryophyllene increased only in the presence of mixed culture. In studies conducted on sucuk, it was determined that autochthonous strains were effective on many terpenes [11,38]. On the other hand, Kaban et al. [12] reported that autochthonous strains were effective only on α-terpineol and camphene, and α-terpineol was reported to increase in the presence of *L. plantarum*. This result could be due to the biotransformation of terpenes by microorganisms [12,50]. On the other hand, it was found that the contents of some terpenes in sausages containing probiotics were lower than in the control group [51].

**Table 5 foods-13-03839-t005:** Overall effect of ripening rate, fat type and starter culture on terpenes of sucuk (mean ± standard deviation) (Au×10^6^).

Compounds	KI	RI	Ripening Rate		Fat Type			Starter Culture			
			Slow	Fast	BF	STF	BF+STF	Control	Lp	Sx	Lp+Sx
Terpenes											
α–thujene	934	b	2.78 ± 4.11 ^a^	1.61 ± 1.83 ^b^	2.07 ± 3.20	2.42 ± 3.55	2.09 ± 2.97	1.79 ± 2.07 ^b^	1.78 ± 1.99 ^b^	1.70 ± 1.99 ^b^	3.50 ± 5.29 ^a^
α-pinene	939	b	16.65 ± 25.78 ^a^	5.43 ± 3.96 ^b^	13.60 ± 25.80	11.53 ± 17.45	7.98 ± 11.73	6.93 ± 10.36 ^b^	5.06 ± 2.95 ^b^	5.98 ± 4.31 ^b^	26.18 ± 32.54 ^a^
Camphene	958	b	0.42 ± 1.06	0.25 ± 0.29	0.26 ± 0.45	0.51 ± 1.23	0.23 ± 0.28	0.28 ± 0.51	0.26 ± 0.32	0.26 ± 0.32	0.53 ± 1.39
Sabinene	971	b	4.03 ± 5.76 ^a^	1.62 ± 2.43 ^b^	2.93 ± 5.25	3.67 ± 5.15	1.88 ± 2.79	3.11 ± 5.66	1.65 ± 1.73	2.56 ± 3.26	3.97 ± 6.04
β-pinene	987	b	15.97 ± 28.77	10.65 ± 8.29	16.63 ± 24.69	14.62 ± 25.99	8.68 ± 7.41	14.72 ± 22.79 ^ab^	7.84 ± 8.05 ^b^	10.41 ± 7.16 ^b^	20.25 ± 33.47 ^a^
β-myrcene	1005	b	84.04 ± 146.97 ^a^	16.86 ± 15.77 ^b^	56.65 ± 123.77	56.36 ± 101.34	38.34 ± 103.20	16.05 ± 14.31 ^b^	24.17 ± 18.01 ^b^	19.59 ± 13.59 ^b^	141.99 ± 191.64 ^a^
α-phellandrene	1022	b	26.36 ± 44.95 ^a^	8.33 ± 8.89 ^b^	21.37 ± 48.75	19.27 ± 27.15	11.40 ± 15.84	9.15 ± 8.54 ^b^	10.04 ± 6.45 ^b^	9.91 ± 8.85 ^b^	40.29 ± 60.65 ^a^
3-carene	1026	b	48.82 ± 76.37 ^a^	13.81 ± 12.66 ^b^	42.45 ± 80.09 ^a^	34.19 ± 52.53 ^a^	17.30 ± 21.97 ^b^	23.09 ± 2.90 ^b^	15.63 ± 12.24 ^b^	19.09 ± 13.89 ^b^	67.44 ± 102.63 ^a^
α-terpinene	1030	b	5.00 ± 9.42 ^a^	2.28 ± 3.68 ^b^	3.50 ± 7.13	5.01 ± 9.37	2.41 ± 4.24	3.08 ± 7.22 ^b^	2.69 ± 3.73 ^b^	1.80 ± 2.13 ^b^	6.99 ± 11.31 ^a^
D-Limonene	1043	b	119.92 ± 183.63 ^a^	28.50 ± 28.32 ^b^	86.89 ± 166.40 ^a^	87.20 ± 146.23 ^a^	48.54 ± 92.04 ^b^	38.53 ± 28.03 ^b^	36.22 ± 33.31 ^b^	35.68 ± 28.87 ^b^	186.41 ± 242.14 ^a^
β-phellandrene	1065	b	13.13 ± 22.57 ^a^	2.96 ± 3.14 ^b^	6.37 ± 12.71	10.69 ± 21.05	7.08 ± 15.75	2.34 ± 2.88 ^b^	4.15 ± 4.15 ^b^	3.48 ± 3.77 ^b^	22.21 ± 29.03 ^a^
β-ocimene	1068	b	0.30 ± 0.55 ^b^	0.85 ± 1.16 ^a^	0.51 ± 0.94	0.63 ± 0.98	0.58 ± 0.93	0.45 ± 0.98 ^ab^	0.80 ± 1.10 ^a^	0.33 ± 0.61 ^b^	0.71 ± 0.98 ^a^
Eucalyptol	1070	b	0.93 ± 2.77	1.21 ± 1.46	1.20 ± 3.16	1.25 ± 1.79	0.77 ± 1.24	1.01 ± 1.41	1.41 ± 3.63	1.43 ± 1.88	0.44 ± 0.74
γ-terpinene	1072	b	93.55 ± 131.07 ^a^	30.13 ± 24.56 ^b^	63.03 ± 84.18 ^ab^	76.09 ± 124.52 ^a^	46.41 ± 83.07 ^b^	35.90 ± 32.55 ^b^	34.73 ± 27.76 ^b^	32.63 ± 21.16 ^b^	144.09 ± 169.26 ^a^
α-terpinolene	1095	b	1.87 ± 4.75	1.57 ± 1.62	1.62 ± 2.60	2.09 ± 5.40	1.45 ± 1.37	1.05 ± 1.28	2.06 ± 2.65	1.32 ± 1.41	2.45 ± 6.25
Linalool	1142	a	29.09 ± 47.39 ^a^	8.46 ± 6.83 ^b^	19.96 ± 35.62	21.47 ± 39.08	14.89 ± 31.13	10.79 ± 8.20 ^b^	9.43 ± 7.43 ^b^	9.09 ± 5.28 ^b^	45.79 ± 62.73 ^a^
4-terpinenol	1220	b	2.75 ± 10.52	1.06 ± 1.36	1.00 ± 1.83	1.71 ± 3.70	3.00 ± 12.38	3.28 ± 13.91	0.86 ± 0.66	0.78 ± 0.81	2.70 ± 5.63
α–terpineol	1252	b	2.13 ± 4.35 ^a^	0.55 ± 0.68 ^b^	1.04 ± 2.13	1.78 ± 4.20	1.20 ± 2.94	0.37 ± 0.51 ^b^	0.61 ± 0.51 ^b^	0.47 ± 0.51 ^b^	3.91 ± 5.65 ^a^
4-carene	1356	b	0.00 ± 0.00 ^b^	0.17 ± 0.41 ^a^	0.06 ± 0.29	0.09 ± 0.33	0.11 ± 0.30	0.07 ± 0.31 ^ab^	0.16 ± 0.43 ^a^	0.11 ± 0.00 ^ab^	0.00 ± 0.00 ^b^
Cumic alcohol	1371	b	11.11 ± 17.73 ^a^	4.32 ± 4.54 ^b^	6.34 ± 12.72	9.52 ± 15.64	7.28 ± 11.39	0.60 ± 1.13 ^c^	5.27 ± 3.18 ^b^	3.00 ± 3.16 ^bc^	21.97 ± 20.39 ^a^
Eugenol	1436	b	0.05 ± 0.13 ^b^	0.51 ± 0.42 ^a^	0.31 ± 0.41	0.26 ± 0.36	0.27 ± 0.40	0.21 ± 0.33 ^b^	0.32 ± 0.47 ^ab^	0.22 ± 0.27 ^b^	0.37 ± 0.43 ^a^
Copaene	1447	b	3.55 ± 5.14 ^a^	1.31 ± 0.67 ^b^	2.41 ± 3.37	2.96 ± 4.84	1.92 ± 3.00	1.10 ± 0.86 ^b^	1.52 ± 0.76 ^b^	1.27 ± 0.58 ^b^	5.83 ± 6.48 ^a^
β–elemene	1453	b	0.04 ± 0.14 ^b^	0.12 ± 0.21 ^a^	0.07 ± 0.18	0.11 ± 0.20	0.07 ± 0.17	0.10 ± 0.21	0.04 ± 0.14	0.07 ± 0.19	0.11 ± 0.20
Iso-caryophyllene	1447	c	4.30 ± 6.88 ^a^	1.44 ± 1.24 ^b^	2.65 ± 5.09	3.43 ± 6.40	2.54 ± 3.57	1.52 ± 1.06 ^b^	1.67 ± 1.36 ^b^	1.38 ± 0.91 ^b^	6.92 ± 0.01 ^a^
Caryophyllene	1490	b	29.58 ± 45.23 ^a^	11.70 ± 5.27 ^b^	24.64 ± 36.25 ^a^	23.69 ± 39.99 ^a^	13.59 ± 19.69 ^b^	12.87 ± 8.23 ^b^	12.78 ± 5.03 ^b^	10.33 ± 4.89 ^b^	46.59 ± 59.08 ^a^
α-caryophyllene	1504	b	0.55 ± 1.08	0.43 ± 0.26	0.33 ± 0.36 ^b^	0.70 ± 1.08 ^a^	0.46 ± 0.71 ^b^	0.17 ± 0.22 ^b^	0.34 ± 0.31 ^b^	0.34 ± 0.15 ^b^	1.13 ± 1.33 ^a^

BF: beef fat; STF: sheep tail fat; Lp: *L. plantarum* GM77; Sx: *S. xylosus* GM92; KI: Kovats index calculated for DB-624 column installed on GC/MS; RI: reliability of identification; a: mass spectrum and retention time identical with authentic sample; b: mass spectrum and Kovats index from literature in accordance; c: tentative identification by mass spectrum. ^a–c^: Means marked with different letters in same row in same factor are statistically different (*p* < 0.05).

### 3.10. Results of Heat Map

A cluster analysis of a heat map showing the relationship between ripening rate and volatile compounds (a) and between ripening rate and chemical groups of volatile compounds (b) is shown in Figure 1. As can be shown in Figure 1a, slow and fast ripening were separated for two clusters, and it was observed that volatile compounds were generally more concentrated in slow ripening. In a study conducted on sucuk, it was determined that the general acceptability of slowly ripened sucuk was higher than that of rapidly ripened sucuk in sensory evaluation [13]. Among the chemical groups determined in sucuk, aromatic hydrocarbons, sulfide compounds, terpenes, alcohols and acid groups were found to be more intense in slowly ripened products (Figure 1b). In addition, ketones, aliphatic hydrocarbons and esters were determined to be relatively more abundant in slow ripening than in fast ripening (Figure 1b). This result showed that slowly ripened products were more aromatic.

The cluster analysis of the heat map shows that two main clusters were formed depending on starter culture. The first main cluster contained only *L. plantarum* + *S. xylosus*, while the second cluster was divided into two clusters within itself. In this cluster, the group with *L. plantarum* was separated into groups with *S. xylosus* and control. On the other hand, the group with *L. plantarum* + *S. xylosus* had more abundant volatile compounds (Figure 2a). In addition, aromatic hydrocarbons, alcohols, esters, aldehydes and ketones, especially terpenes and sulfide compounds, were more correlated with the group with *L. plantarum + S. xylosus* than with the other groups (Figure 2b). As a result, the use of mixed culture resulted in more intense volatile compounds.

In terms of fat type, two main clusters were formed and the sheep tail fat group was separated from the beef fat and beef fat+sheep tail fat groups. The groups containing beef fat showed a close correlation to each other. Although a different abundance was observed in a limited number of compounds among the volatile compounds, it was determined that sheep tail fat had a different volatile profile (Figure 3a). In terms of sulfide compounds, especially the sheep tail fat group showed a different feature than the other groups and contained this group of compounds more intensively. On the other hand, the beef fat group contained relatively more aromatic hydrocarbons compared to the other groups (Figure 3b). In the study conducted by Akköse et al. [13], it was also stated that the use of sheep tail fat in sucuk reduced the odor and general acceptability properties.

## 4. Conclusions

Fermented sausages such as sucuk have a complex matrix, and different reactions occur during ripening. The rate and type of these reactions vary depending on internal and external factors. This study was limited to the effects of ripening rate, starter culture and fat type on the volatile compounds of sucuk. Slow ripening increased the abundance of many volatile compounds. Similarly, mixed culture was also effective in many compounds. Many terpene compounds were affected by the use of mixed culture. On the other hand, the effect of the use of different fat types on volatile compounds was limited. However, in the case of sheep tail fat, while sulfide compounds came to the fore, the aromatic hydrocarbon content increased relatively more in the presence of beef fat. It is concluded that slow ripening and the use of mixed cultures were necessary to produce a more aromatic sucuk.

## Figures and Tables

**Figure 1 foods-13-03839-f001:**
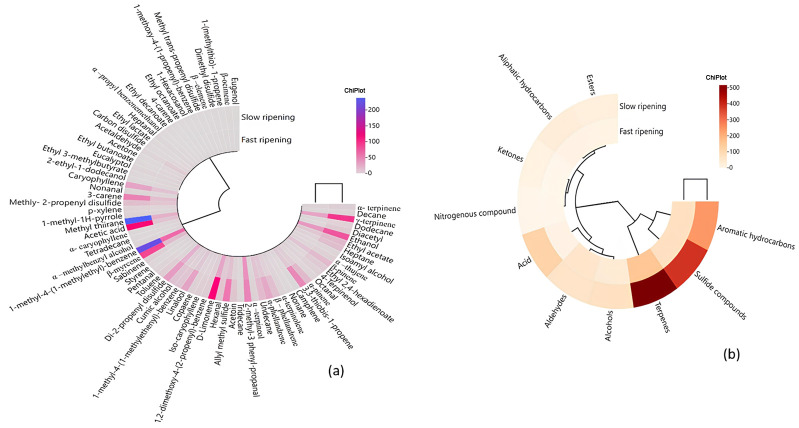
Cluster analysis of heat map showing relationship between ripening rate and volatile compounds (**a**) and between ripening rate and chemical groups of volatile compounds (**b**).

**Figure 2 foods-13-03839-f002:**
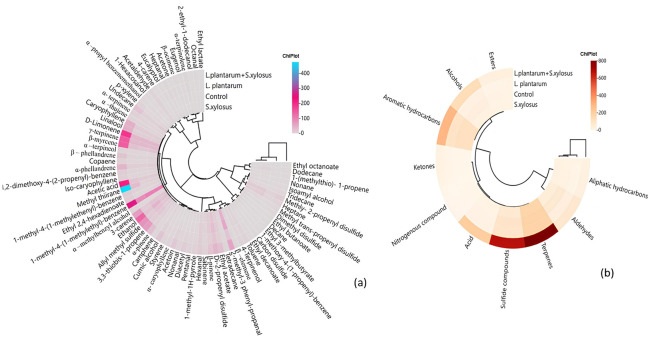
Cluster analysis of heat map showing relationship between starter culture and volatile compounds (**a**) and between starter culture and chemical groups of volatile compounds (**b**).

**Figure 3 foods-13-03839-f003:**
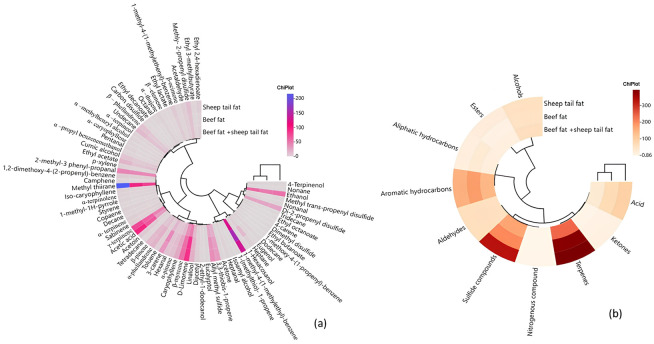
Cluster analysis of heat map showing the relationship between fat type and volatile compounds (**a**) and between fat type and chemical groups of volatile compounds (**b**).

**Table 1 foods-13-03839-t001:** Experimental design.

Slow Ripening				Fast Ripening			
Starter Culture	Type of Fat			Starter Culture	Type of Fat		
Control	BF	STF	BF+STF	Control	BF	STF	BF+STF
*L. plantarum* GM77	BF	STF	BF+STF	*L. plantarum* GM77	BF	STF	BF+STF
*S. xylosus* GM92	BF	STF	BF+STF	*S. xylosus* GM92	BF	STF	BF+STF
*L. plantarum* GM77 *S. xylosus* GM92	BF	STF	BF+STF	*L. plantarum* GM77 *S. xylosus* GM92	BF	STF	BF+STF

BF: beef fat; STF: sheep tail fat.

## Data Availability

The original contributions presented in this study are included in the article. Further inquiries can be directed to the corresponding authors.

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
