# Peer review of "Volatile Compounds of Sucuk, a Dry Fermented Sausage: The Effects of Ripening Rate, Autochthonous Starter Cultures and Fat Type"

_foods, 2024, doi:10.3390/foods13233839_

Round 1

Reviewer 1 Report

Comments and Suggestions for Authors

The aim of the manuscript was to determine the effects of ripening rate (slow or fast), usage autochthonous starter cultures and type of fat on the volatile compounds of Turkish dry fermented sausage. The title of manuscript corresponds to the content, the aim is correctly formulated.

The conducted research is interesting, but in my opinion it lacks a sensory evaluation of the products, which would allow for an in-depth discussion of the obtained results on volatile compounds.

Moreover, the article contains some inaccuracies that need to be improved.

Line 22 - keyword "fat" is unnecessary in my opinion.

Lines 80-81 - the Authors wrote that „Nitrate ......was added to the batters for the fast ripening, and nitrite .... for the slow ripening” - why in sausage technology using fast fermentation, nitrate was used, which requires time to be reduced first to nitrite and then to nitric oxide? When discussing the results (lines 133- 135), the Authors refer to studies in which nitrite in fast ripening and nitrate in slow ripening were used. Please explain.

Lines 147-154 - please indicate the correct table where the discussed results are located (this is not table 1)

Lines 160 -  please indicate the correct table where the discussed results are located (this is not table 1)

Tables 2-5 - please replace commas with dots in the data

Conclusion does not fulfill its role, it is a repetition of results. Needs improvement.

Author Response

The aim of the manuscript was to determine the effects of ripening rate (slow or fast), usage autochthonous starter cultures and type of fat on the volatile compounds of Turkish dry fermented sausage. The title of manuscript corresponds to the content, the aim is correctly formulated.

The conducted research is interesting, but in my opinion it lacks a sensory evaluation of the products, which would allow for an in-depth discussion of the obtained results on volatile compounds.

Moreover, the article contains some inaccuracies that need to be improved.

Line 22 - keyword "fat" is unnecessary in my opinion.

-Thank you for your comment.  Fat was removed.

Lines 80-81 - the Authors wrote that „Nitrate ......was added to the batters for the fast ripening, and nitrite .... for the slow ripening” - why in sausage technology using fast fermentation, nitrate was used, which requires time to be reduced first to nitrite and then to nitric oxide? When discussing the results (lines 133- 135), the Authors refer to studies in which nitrite in fast ripening and nitrate in slow ripening were used. Please explain.

-First of all, thank you very much for your attention. It is typing error.  We corrected it. Due to the production process, nitrite is used for fast ripening and nitrate is used for slow ripening.

Lines 147-154 - please indicate the correct table where the discussed results are located (this is not table 1)

-Thank you. Corrected.

Lines 160 -  please indicate the correct table where the discussed results are located (this is not table 1)

-Thank you. Corrected.

Tables 2-5 - please replace commas with dots in the data

-Thank you. Changed.

Conclusion does not fulfill its role, it is a repetition of results. Needs improvement.

-Thank you. Revised.

Reviewer 2 Report

Comments and Suggestions for Authors

Overall, this article is well-written, well-organized and easy to understand, and it is a part of comprehensive research on sucuk (Akköse, A.; OÄŸraÅŸ, Åž. Åž.; Kaya, M.; Kaban, G. Microbiological, physicochemical and sensorial changes during the ripening 370 of sucuk, a traditional Turkish dry-fermented sausage: Effects of autochthonous strains, sheep tail fat and ripening rate. Fermentation, 2023, 9(6), 558).
Sucuk is a meat product made from ruminant meat and fat, mainly from beef meat and fat. This product is characterized by a high fat content. The amount and type of fat have a great influence on the sensory quality of dry fermented sausages.
In this work, part of the beef fat was replaced with sheep fat.  This is particularly interesting because it is a traditional product that is also produced in rural areas. In addition, the fat-tailed sheep is a general type of domestic sheep.
The use of sheep fat is the most significant scientific contribution of the work. The topic of work is original and unexplored so far.
In addition, it would be also crucial that sucuk was made only from sheep tissues (meat and fat) and its quality compared to others that were showed.
Conclusions are consistent with the data and their analysis.
The chosen references are appropriate and ensure credibility and reliability.
Keywords Changed lactic acid bacteria’ with ‘starter cultures’

Author Response

We would like to thank the reviewer for the comments.

Kind regards,

Reviewer #2:

Overall, this article is well-written, well-organized and easy to understand, and it is a part of comprehensive research on sucuk (Akköse, A.; OÄŸraÅŸ, Åž. Åž.; Kaya, M.; Kaban, G. Microbiological, physicochemical and sensorial changes during the ripening 370 of sucuk, a traditional Turkish dry-fermented sausage: Effects of autochthonous strains, sheep tail fat and ripening rate. Fermentation, 2023, 9(6), 558).
Sucuk is a meat product made from ruminant meat and fat, mainly from beef meat and fat. This product is characterized by a high fat content. The amount and type of fat have a great influence on the sensory quality of dry fermented sausages.

-Thank you very much for your comments.

In this work, part of the beef fat was replaced with sheep fat.  This is particularly interesting because it is a traditional product that is also produced in rural areas. In addition, the fat-tailed sheep is a general type of domestic sheep.
The use of sheep fat is the most significant scientific contribution of the work. The topic of work is original and unexplored so far.
In addition, it would be also crucial that sucuk was made only from sheep tissues (meat and fat) and its quality compared to others that were showed.

-Thank you very much for your comments.

Conclusions are consistent with the data and their analysis.
The chosen references are appropriate and ensure credibility and reliability.

-Thank you very much for your comments.

Keywords Changed ’lactic acid bacteria’ with ‘starter cultures’

-Thank you very much for your comments. Revised.

Reviewer 3 Report

Comments and Suggestions for Authors

In this manuscript, the author summarized and discussed the effects of volatile substances in fermented sausages on consumer preferences. On this basis, they further studied the effects of different fermentation time, starter culture and different fat composition on volatile compounds in fermented Turkish sausages, and determined the values of various volatile compounds in fermented sausages of different treatment groups. This work provides new insight and opinion on the selection of fermented sausages with different ingredients and treatments. The manuscript is well-organized and clearly stated. I would suggest accepting it after the following minor concerns are addressed.

1.       It is suggested that the introduction section introduce in more detail the cultural and economic importance of sucuk and the influence of volatile compounds on food flavor.

2.       It is suggested to discuss the significance of the results in depth in the final summary, and clearly point out the limitations of the study, such as sample size and experimental conditions, so as to provide the direction for future research.

Author Response

We would like to thank the reviewer for the comments. Our responses to the reviewer are attached to the text in red color. 

Kind regards,

Reviewer #3:

In this manuscript, the author summarized and discussed the effects of volatile substances in fermented sausages on consumer preferences. On this basis, they further studied the effects of different fermentation time, starter culture and different fat composition on volatile compounds in fermented Turkish sausages, and determined the values of various volatile compounds in fermented sausages of different treatment groups. This work provides new insight and opinion on the selection of fermented sausages with different ingredients and treatments. The manuscript is well-organized and clearly stated. I would suggest accepting it after the following minor concerns are addressed.

  1. It is suggested that the introduction section introduce in more detail the cultural and economic importance of sucuk and the influence of volatile compounds on food flavor.

- Thank you for your comments. Revised.

  1. It is suggested to discuss the significance of the results in depth in the final summary, and clearly point out the limitations of the study, such as sample size and experimental conditions, so as to provide the direction for future research.

- Thank you for your comments. Added to the conclusion.

Round 2

Reviewer 1 Report

Comments and Suggestions for Authors

'The manuscript has been sufficiently improved to warrant publication in Foods.